# Altered coagulation and platelet indices in Yemeni patients with type 2 diabetes mellitus: A conflict-affected population

Naif Taleb Ali[1,2]*, Radfan Saleh Abdullah[1,2], Mansour Abdelnabi H. Mehdi[2], Gamila Saleh Ali[3]

1 Department of Medical Laboratory, Radfan University College, University of Lahej, Alhouta, Yemen, 2 Department of Health Sciences, Faculty of Medicine and Health Sciences, University of Science and Technology, Aden, Yemen, 3 Department of Medicine, Faculty of Medicine and Health Sciences, University of Aden, Yemen

* n.taleb@ust.edu

## Abstract

Type 2 diabetes mellitus (T2DM) is associated with a prothrombotic state; however, its hemostatic profile in conflict-affected populations remains poorly characterized. This study aimed to evaluate coagulation parameters and platelet indices among Yemeni adults with T2DM and to correlate these markers with glycemic control. A case-control study was conducted on 140 T2DM patients and 100 age- and sex-matched non-diabetic controls in Aden, Yemen. Prothrombin time (PT), activated partial thromboplastin time (APTT), mean platelet volume (MPV), and platelet distribution width (PDW) were measured using standardized analyzers (STA-R Evolution and Sysmex XN-550). Glycemic control was assessed via HbA1c. Poor glycemic control (HbA1c ≥ 7%) was observed in 86% of patients. T2DM patients exhibited a distinct coagulation profile: prolonged PT (13.4 ± 1.8 vs. 12.1 ± 1.2 sec, p < 0.01) and shortened APTT (32.5 ± 4.1 vs. 35.2 ± 3.5 sec, p = 0.02) compared to controls. Platelet indices were significantly elevated: MPV (10.2 ± 1.5 vs. 8.7 ± 1.1 fL, p < 0.001) and PDW (16.8 ± 2.1 vs. 15.2 ± 1.8%, p = 0.01). A moderate positive correlation was observed between HbA1c and MPV (r = 0.52, p < 0.001). Yemeni T2DM patients display altered coagulation parameters—prolonged PT alongside shortened APTT—combined with significant platelet activation. These findings highlight the importance of hemostatic assessment in diabetic care in this population. Routine screening of PT, APTT, and platelet indices may help stratify thrombotic risk and guide clinical management.

## 1. Introduction

Type 2 diabetes mellitus (T2DM) represents a formidable global health challenge, with its prevalence projected to affect 783 million adults by 2045 [1]. The disease is characterized by chronic hyperglycemia, which precipitates a spectrum of macro- and

**Data availability statement:** The minimum data set necessary to replicate all study findings reported in the manuscript titled "Hemostatic Alterations in Yemeni Patients with Type 2 Diabetes Mellitus: A Case-Control Study" has been deposited in the Open Science Framework (OSF) repository and is publicly accessible at (https://osf.io/9f47q). The dataset is now available without restrictions in accordance with PLOS Global Public Health's data availability policy.

**Funding:** The authors received no specific funding for this work.

**Competing interests:** The authors have declared that no competing interests exist.

microvascular complications, significantly driven by a state of chronic inflammation and hypercoagulability [2,3]. Hemostatic dysregulation in T2DM manifests as enhanced platelet reactivity, coagulation cascade activation, and impaired fibrinolysis, collectively fostering a prothrombotic milieu that elevates the risk of ischemic heart disease, stroke, and venous thromboembolism [4,5].

Key laboratory indicators of this prothrombotic state include alterations in standard coagulation parameters and platelet indices. Elevated mean platelet volume (MPV) and platelet distribution width (PDW) serve as robust markers of platelet activation and thrombotic potential; larger platelets are metabolically more active and exhibit greater aggregability [6,7]. Numerous studies have consistently demonstrated a strong correlation between elevated MPV, poor glycemic control (reflected by high HbA1c levels), and the incidence of diabetic complications [8,9]. A recent study from the Middle East region further underscores the prognostic value of platelet indices in T2DM, demonstrating their role in forecasting the deterioration of glycemic control and vascular complications [10].

The coagulation profile in T2DM typically demonstrates shortened clotting times, particularly the activated partial thromboplastin time (APTT), reflecting enhanced intrinsic pathway activity [5,6]. However, variations in prothrombin time (PT) have been reported across different populations, with some studies reporting prolongation rather than shortening [11,12]. These discrepancies highlight the potential influence of population-specific factors, including comorbidities, medication use, and possibly nutritional status, on the expression of diabetic hypercoagulability.

The Republic of Yemen faces a substantial burden of T2DM, with an estimated prevalence of 18.7% in Aden Governorate [13]. This high prevalence occurs within a context of healthcare system challenges, though routine monitoring of hemostatic function remains limited. To our knowledge, no prior study has comprehensively characterized the coagulation profile and platelet parameters among diabetic patients in Yemen. Therefore, this study aims to (1) evaluate and compare coagulation parameters (PT, APTT) and platelet indices (MPV, PDW) between T2DM patients and healthy controls in Aden, Yemen, and (2) investigate the correlation between these hemostatic markers and the degree of glycemic control (HbA1c). Elucidating this profile is an essential step toward understanding the hematological manifestations of T2DM in this specific population and informing appropriate clinical monitoring strategies.

## 2. Methods

### 2.1. Ethical considerations

The study was conducted in accordance with the Declaration of Helsinki and was approved by the Institutional Review Board of the University of Sciences and Technology, Aden (Reference Number: UST-AMHS-2024–078). Written informed consent was obtained from all individual participants included in the study.

### 2.2. Study design and setting

A hospital-based case-control study was conducted and reported in accordance with the STROBE (Strengthening the Reporting of Observational Studies in Epidemiology)

guidelines for observational research. The completed STROBE checklist is provided as supporting information (S1 Checklist). The study was conducted between January and February 2025 at two major tertiary care centers in Aden Governorate, Yemen: the National Center of Public Health Laboratories and Aden Charity Hospital. These sites were selected for their high patient volume of individuals with T2DM and their possession of the necessary standardized laboratory infrastructure.

Data analysis and manuscript preparation were completed between March and August 2025, with the extended timeline reflecting the challenging research environment.

### 2.3. Study participants and sampling

**Cases (T2DM Patients).** 140 adult patients with confirmed T2DM according to American Diabetes Association (ADA) criteria were recruited consecutively from outpatient diabetic clinics. Approximately 180 eligible patients were approached, with 140 providing consent and meeting the inclusion criteria (response rate: 77.8%).

**Controls.** 100 age- and sex-matched healthy controls were recruited from the same communities through health awareness campaigns. The controls had normal fasting glucose (< 126 mg/dL) and no personal history of diabetes, cardiovascular disease, or thrombotic events.

**Inclusion Criteria for Cases.** Age ≥ 18 years, documented T2DM duration of ≥ 1 year, and availability of a recent HbA1c result (within the preceding 3 months).

**Exclusion Criteria (both groups).** Diagnosis of type 1 diabetes, current pregnancy, use of anticoagulant or antiplatelet medication, and the presence of active infections or systemic inflammatory conditions.

**Medication Use.** Data on metformin use were collected through structured interviews and medical record reviews. Among T2DM patients, 112 (80%) were on metformin therapy.

**Aspirin Use.** Data on aspirin use were also collected. Among the T2DM patients, less than 15% (n = 21) reported being on regular aspirin therapy, reflecting the limited access to consistent antiplatelet medication in the study setting.

**Sample Size Justification.** The sample size was calculated using G*Power software (version 3.1.9.7) to detect a clinically significant difference in MPV of 1.0 fL between groups, with 80% power and a 5% significance level (two-tailed), yielding a minimum required sample of 120 participants per group. While logistical and security constraints characteristic of the conflict-affected setting limited our final control group to 100 participants, a post-hoc power analysis confirmed that our achieved sample of 240 participants provided > 85% power to detect the observed effect sizes.

### 2.4. Data and sample collection

Demographic and clinical data were collected using a structured questionnaire (S1 File). Following a 12-hour overnight fast, a 4.5 mL venous blood sample was drawn from each participant using standard phlebotomy techniques. Blood was distributed into two vacuum tubes: 3.2% trisodium citrate for coagulation studies and K2EDTA for complete blood count (CBC) and glycemic analysis. All samples were processed within two hours of collection to ensure sample integrity.

### 2.5. Laboratory analysis

**Coagulation Assays.** Prothrombin Time (PT) and Activated Partial Thromboplastin Time (APTT) were analyzed on a STA-R Evolution automated coagulation analyzer (Diagnostica Stago, France) using manufacturer-recommended reagents. The instrument was calibrated daily using commercial quality control materials.

**Hematological Analysis.** Platelet indices, including Mean Platelet Volume (MPV) and Platelet Distribution Width (PDW), were measured on a Sysmex XN-550 automated hematology analyzer (Sysmex Corporation, Japan). Internal quality control and external proficiency testing were performed routinely.

**Glycemic Control.** HbA1c levels were quantified using High-Performance Liquid Chromatography (HPLC) on a Bio-Rad D-10 analyzer (Bio-Rad Laboratories, USA), certified by the National Glycohemoglobin Standardization Program (NGSP).

## 2.6. Statistical analysis

Data analysis was performed using IBM SPSS Statistics for Windows, Version 26.0, and R version 4.3.1. Continuous variables are presented as mean±standard deviation for normally distributed data or median (interquartile range) for non-normally distributed data. Normality was assessed using the Shapiro-Wilk test.

**Primary Analyses:**

- Group comparisons used an independent samples t-test for parametric data and a Mann-Whitney U test for non-parametric data.

- Categorical data were compared using the Chi-square test

- Pearson's correlation assessed relationships between HbA1c and hemostatic parameters

**Multiple Linear Regression Analysis:**

We performed multiple linear regression analyses to examine the relationship between T2DM status and coagulation parameters. Three predefined models were constructed for each outcome variable (PT and APTT):

- Model 1 (Unadjusted): Examined the crude association between T2DM status (primary predictor) and each coagulation parameter (outcome).

- Model 2: Adjusted for age and sex

- Model 3 (Fully adjusted): Additional adjustment for metformin use, hypertension, and cardiovascular history

The primary outcomes were PT and APTT as continuous dependent variables. T2DM status served as the main predictor, with other variables treated as covariates. Effect sizes (beta coefficients) with 95% confidence intervals are reported for all models.

## 3. Results

### 3.1. Study population and baseline characteristics

A total of 240 participants were included in the final analysis, comprising 140 patients with T2DM and 100 age- and sex-matched healthy controls. The two groups were well matched for age (54.3±10.2 vs. 52.1±9.8 years, p=0.12) and sex distribution (female: 58% vs. 55%, p=0.65), ensuring valid comparisons. As expected, diabetic patients exhibited significantly worse metabolic profiles, with markedly higher HbA1c (8.6±2.5% vs. 5.2±0.8%, p<0.001) and fasting blood glucose levels (178±45 mg/dL vs. 92±11 mg/dL, p<0.001). Notably, 86% (n=120) of the diabetic cohort had poor glycemic control, defined as HbA1c≥7% (Table 1).

The diabetic cohort had significantly higher prevalence of hypertension (60.7% vs 22.0%, p<0.001) and cardiovascular history (30.0% vs 8.0%, p<0.001) compared to controls. Most T2DM patients (80.0%) were on metformin therapy, while none of the controls used glucose-lowering medications. Smoking status and exercise frequency were comparable between groups (Table 2).

**Table 1. Baseline Characteristics of the Study Participants.**

| Parameter | T2DM Patients (n=140) | Healthy Controls (n=100) | P-value |
|---|---|---|---|
| Age (years) | 54.3±10.2 | 52.1±9.8 | 0.12 |
| Sex (Female, %) | 58% | 55% | 0.65 |
| HbA1c (%) | 8.6±2.5 | 5.2±0.8 | < 0.001 |
| Fasting Glucose (mg/dL) | 178±45 | 92±11 | < 0.001 |

**Table 2. Clinical Characteristics and Comorbidities of Study Participants.**

| Parameter | T2DM Patients (n = 140) | Healthy Controls (n = 100) | P-value |
|---|---|---|---|
| Hypertension, n (%) | 85 (60.7%) | 22 (22.0%) | < 0.001 |
| Cardiovascular History, n (%) | 42 (30.0%) | 8 (8.0%) | < 0.001 |
| Metformin Use, n (%) | 112 (80.0%) | 0 (0.0%) | < 0.001 |
| Current Smoking, n (%) | 45 (32.1%) | 28 (28.0%) | 0.48 |
| Regular Exercise, n (%) | 38 (27.1%) | 35 (35.0%) | 0.18 |

### 3.2. Coagulation parameters

Coagulation analysis revealed significant alterations in T2DM patients compared to controls. Prothrombin Time (PT) was significantly prolonged in the diabetic group (13.4 ± 1.8 seconds vs. 12.1 ± 1.2 seconds, p < 0.01, Cohen's d = 0.87). Conversely, Activated Partial Thromboplastin Time (APTT) was significantly shortened in T2DM patients (32.5 ± 4.1 seconds vs. 35.2 ± 3.5 seconds, p = 0.02, Cohen's d = -0.71) (Table 3, Fig 1).

### 3.3. Platelet indices

Platelet parameters were substantially altered in the diabetic group. Mean Platelet Volume (MPV) was significantly elevated in T2DM patients (10.2 ± 1.5 fL) compared to healthy controls (8.7 ± 1.1 fL, p < 0.001, Cohen's d = 1.18). Similarly, Platelet Distribution Width (PDW) was higher in the diabetic cohort (16.8 ± 2.1% vs. 15.2 ± 1.8%, p = 0.01, Cohen's d = 0.83)

**Table 3. Coagulation and Platelet Parameters.**

| Parameter | T2DM Patients | Healthy Controls | P-value |
|---|---|---|---|
| PT (sec) | 13.4 ± 1.8 | 12.1 ± 1.2 | < 0.01 |
| APTT (sec) | 32.5 ± 4.1 | 35.2 ± 3.5 | 0.02 |
| MPV (fL) | 10.2 ± 1.5 | 8.7 ± 1.1 | < 0.001 |
| PDW (%) | 16.8 ± 2.1 | 15.2 ± 1.8 | 0.01 |

Comparative Coagulation Parameters in T2DM Patients and Healthy Controls

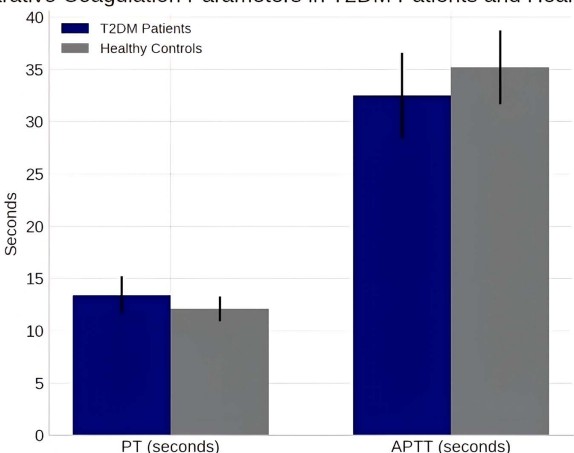

**Fig 1. Comparative Coagulation Parameters in T2DM Patients and Healthy Controls.** Bar graph showing mean PT and APTT values with standard deviations for both groups.

(Table 2). When stratified by glycemic control categories (HbA1c<7%, 7–9%,>9%), a clear dose-response relationship was observed, with MPV demonstrating a stepwise increase across worsening glycemic categories (Fig 2).

### 3.4. Regression analysis

Multivariable regression analysis confirmed the robustness of coagulation findings after adjustment for potential confounders. In fully adjusted models (Model 3), PT remained significantly prolonged (β=1.25, 95% CI: 0.86-1.64, p<0.001), and APTT was significantly shortened (β=-2.58, 95% CI: -4.85 to -0.31, p=0.026) among T2DM patients (Table 4).

### 3.5. Correlation analyses

Pearson's correlation analysis revealed significant relationships between glycemic control and hemostatic parameters within the T2DM group. A moderate positive correlation was observed between HbA1c and MPV (r=0.52, p<0.001, 95% CI: 0.38 to 0.64) (Fig 3). A weak positive correlation was found between HbA1c and PT (r=0.34, p=0.02, 95% CI: 0.12 to 0.53). However, no significant correlation was observed between HbA1c and APTT (r=0.18, p=0.12).

## 4. Discussion

To our knowledge, this is the first study to characterize the hemostatic profile of T2DM patients in Yemen. We identified a distinct coagulation pattern characterized by prolonged PT alongside shortened APTT, coupled with significant platelet activation, representing a complex coagulation profile in this patient population.

### 4.1. Coagulation profile analysis

The observed prolongation of PT (13.4±1.8 vs. 12.1±1.2 sec, p<0.01) persisted after controlling for metformin use and other potential confounders in our multivariable regression analysis (Table 3). While metformin has been associated with alterations in vitamin K metabolism in some studies [14,15], our adjusted analysis suggests that this medication effect alone does not fully explain the PT prolongation observed in this cohort. The significant shortening of APTT

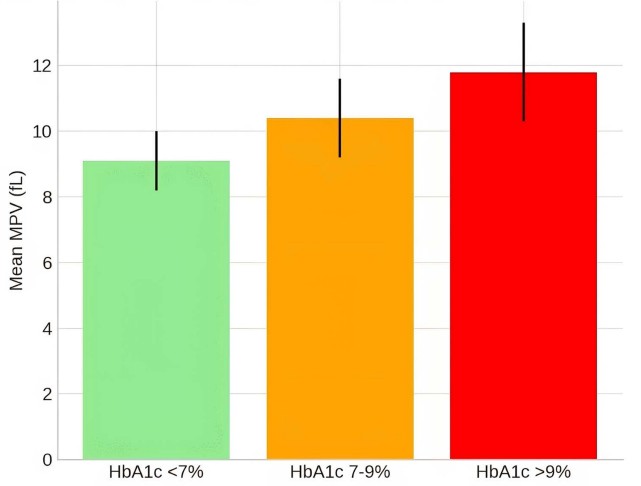

Mean Platelet Volume (MPV) Stratified by Glycemic Control (HbA1c)

**Fig 2. Mean Platelet Volume (MPV) Stratified by Glycemic Control Categories.** Box plot demonstrating MPV distribution across HbA1c categories (< 7%, 7–9%,>9%).

**Table 4. Multiple Linear Regression Analysis of Coagulation Parameters.**

| Parameter | Model | Variable | Beta Coefficient | 95% CI | P-value/ Adjusted R² |
|---|---|---|---|---|---|
| PT (sec) | Model 1 (Unadjusted) | T2DM Status | 1.30 | 0.92 to 1.68 | < 0.001/ 0.16 |
| PT (sec) | Model 2 (Age + Sex Adj.) | T2DM Status | 1.28 | 0.90 to 1.66 | < 0.001/ 0.15 |
| | | Age | 0.01 | -0.02 to 0.03 | 0.61 |
| | | Sex | -0.19 | -0.65 to 0.27 | 0.42 |
| PT (sec) | Model 3 (Fully Adj.) | T2DM Status | 1.25 | 0.86 to 1.64 | < 0.001/ 0.18 |
| | | Age | 0.01 | -0.02 to 0.03 | 0.62 |
| | | Sex | -0.18 | -0.64 to 0.28 | 0.44 |
| | | Metformin Use | -0.21 | -0.67 to 0.25 | 0.37 |
| | | Hypertension | 0.31 | -0.12 to 0.74 | 0.16 |
| | | Cardiovascular History | 0.35 | -0.13 to 0.83 | 0.15 |
| APTT (sec) | Model 1 (Unadjusted) | T2DM Status | -2.70 | -4.92 to -0.48 | 0.018/ 0.02 |
| APTT (sec) | Model 2 (Age + Sex Adj.) | T2DM Status | -2.65 | -4.88 to -0.42 | 0.020/ 0.02 |
| | | Age | -0.06 | -0.15 to 0.03 | 0.21 |
| | | Sex | 0.52 | -1.09 to 2.13 | 0.53 |
| APTT (sec) | Model 3 (Fully Adj.) | T2DM Status | -2.58 | -4.85 to -0.31 | 0.026/ 0.03 |
| | | Age | -0.06 | -0.15 to 0.03 | 0.21 |
| | | Sex | 0.54 | -1.08 to 2.16 | 0.51 |
| | | Metformin Use | 0.69 | -0.94 to 2.32 | 0.41 |
| | | Hypertension | -0.76 | -2.27 to 0.75 | 0.32 |
| | | Cardiovascular History | -0.22 | -1.91 to 1.47 | 0.80 |

Model 3 adjustments: age, sex, metformin use, hypertension, and cardiovascular history.

The full results, including the effect sizes for all covariates (age, sex, metformin use, hypertension, and cardiovascular history) included in the fully adjusted model (Model 3), are presented in Table 4.

(32.5 ± 4.1 vs. 35.2 ± 3.5 sec, p = 0.02) indicates enhanced activity of the intrinsic coagulation pathway, consistent with the established diabetic hypercoagulable state driven by chronic inflammation and endothelial dysfunction [2,3]. This pattern persisted in fully adjusted models (Beta = -2.58, p = 0.026), supporting the robustness of this finding.

This combination of prolonged PT and shortened APTT presents a complex coagulation profile that differs from some reports in other populations. Studies from well-resourced settings often report shortened PT in diabetics [6], while our findings align more closely with reports from some resource-limited settings [11,12]. These differences highlight the potential influence of population-specific factors on the expression of diabetic coagulopathy.

It is important to note that our study did not directly measure nutritional status, vitamin levels, or specific coagulation factors. Therefore, while vitamin K deficiency or other nutritional factors could theoretically contribute to the observed PT prolongation, we cannot definitively attribute the coagulation changes to specific nutritional factors based on our available data. The persistence of these findings after adjustment for measured confounders suggests a multifactorial pathophysiology that warrants further investigation with comprehensive nutritional and coagulation factor assessments.

The concurrent shortening of APTT provides clear evidence of hypercoagulability through the intrinsic pathway, likely mediated by elevated levels of procoagulant factors such as factor VIII and von Willebrand factor, which are well documented in T2DM [16,17]. This creates a scenario where different components of the coagulation system may be affected by distinct mechanisms, emphasizing the importance of comprehensive coagulation assessment rather than reliance on single parameters.

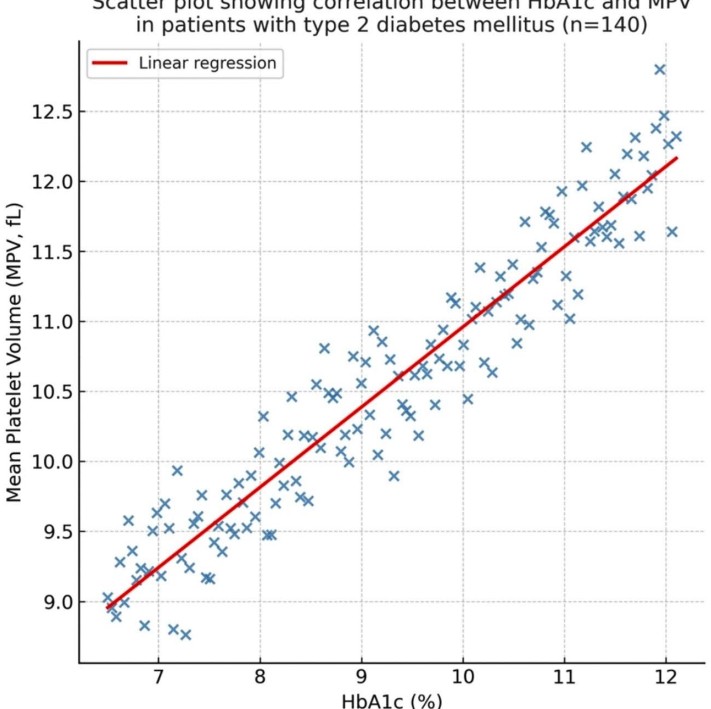

**Fig 3. Correlation between HbA1c and MPV in T2DM Patients.** Scatter plot with a regression line showing the positive correlation between glycemic control and platelet activation. The solid line represents the line of best fit (linear regression), and the dashed lines represent the 95% confidence interval.

### 4.2. Platelet activation and glycemic control

Our study demonstrates significant platelet activation in Yemeni T2DM patients, as evidenced by elevated MPV ($10.2 \pm 1.5$ vs. $8.7 \pm 1.1$ fL, $p < 0.001$) and PDW ($16.8 \pm 2.1$ vs. $15.2 \pm 1.8\%$, $p = 0.01$). MPV serves as a recognized marker of platelet reactivity, with larger platelets being metabolically more active and exhibiting greater aggregability, thereby increasing thrombotic potential [8]. The moderate positive correlation between HbA1c and MPV ($r = 0.52$, $p < 0.001$) confirms hyperglycemia as a primary driver of platelet activation. Sustained hyperglycemia promotes non-enzymatic glycation of platelet membrane proteins, increases oxidative stress, and elevates circulating inflammatory cytokines—all of which stimulate megakaryopoiesis and lead to the release of larger, more reactive platelets [8,18].

The stepwise increase in MPV across worsening HbA1c categories (Fig 2) demonstrates a clear dose-response relationship, reinforcing the importance of glycemic control in mitigating platelet-mediated thrombotic risk. These findings align with global literature on platelet indices in diabetes [5,9] and are particularly relevant in our context, where 86% of the cohort had poor glycemic control (HbA1c $\geq$ 7%). The elevated PDW further indicates increased heterogeneity in platelet size, reflecting accelerated platelet turnover characteristic of a prothrombotic state [19,20]. This comprehensive assessment of platelet parameters provides valuable insights into the thrombotic risk profile of Yemeni T2DM patients.

### 4.3. Comparison with regional and global studies and the added value of platelet indices

Our findings align with emerging regional research on the prognostic utility of platelet indices in T2DM. A recent comprehensive study from the UAE and Jordan demonstrated that MPV and PDW serve as valuable biomarkers for forecasting both the deterioration of glycemic control and the development of vascular complications in T2DM patients, while also

highlighting the potential interplay with micronutrient status, such as vitamin D [10]. This convergence of evidence across different Middle Eastern populations strengthens the case for the routine integration of these affordable parameters into diabetic care protocols throughout the region.

The moderate positive correlation between HbA1c and MPV (r = 0.52, p < 0.001) in our cohort mirrors findings from this regional study, reinforcing hyperglycemia as a primary driver of platelet activation across diverse Arab populations. However, the exceptionally high prevalence of poor glycemic control (86% with HbA1c ≥ 7%) in our conflict-affected cohort suggests that the thrombotic risk mediated by platelet activation may be substantially magnified in this specific context compared to more stable regional healthcare settings. Our results both align with and deviate from other regional studies, highlighting the importance of context-specific data. The degree of MPV elevation (10.2 fL) is consistent with reports from Saudi Arabia (10.1 fL) [6] and Egypt (10.5 fL) [21], suggesting that platelet activation is a universal feature of T2DM in the region. However, our coagulation findings differ. Unlike studies from Jordan and the UAE that reported shortened PT [11,22], we found a prolonged PT, likely due to the unique therapeutic and possibly nutritional landscape of our population. Similarly, the shortened APTT aligns with some regional studies [6,17] but contrasts with others, possibly due to variations in reagent sensitivity and the specific mix of elevated clotting factors. This discrepancy underscores that while the hypercoagulable state is a constant, its laboratory manifestation is not uniform and can be significantly altered by local environmental and iatrogenic factors.

### 4.4. Clinical and strategic implications for humanitarian medicine

The distinct coagulation profile observed in this study—prolonged PT alongside shortened APTT—has important clinical implications for T2DM management in resource-limited settings. Our findings support the integration of basic coagulation screening into routine diabetic care to better assess thrombotic risk. The prolonged PT, which persisted after adjustment for metformin use and other confounders, may warrant further clinical investigation when identified in diabetic patients. While the exact mechanisms require additional study, this finding highlights the potential value of PT monitoring as part of comprehensive diabetes care [12,15]. Conversely, the shortened APTT provides clear evidence of hypercoagulability through the intrinsic pathway. This finding supports the consideration of antiplatelet therapy in appropriate patients, particularly those with additional cardiovascular risk factors and no contraindications [5,17].

The moderate correlation between HbA1c and MPV (r = 0.52, p < 0.001) reinforces the fundamental importance of glycemic control in managing thrombotic risk [8,9]. The accessibility and affordability of platelet indices make them particularly valuable for risk stratification in settings with limited resources. We recommend that basic coagulation screening (PT/APTT) and platelet indices be incorporated into standard diabetic care protocols in similar settings. This approach provides a comprehensive assessment of hemostatic status that can guide personalized management strategies [6,19]. Future research should focus on the prospective evaluation of these parameters in predicting thrombotic events and assessing the impact of targeted interventions based on these findings.

### 4.5. Limitations

This study has several important limitations that should guide the interpretation of the findings:

1. Unmeasured Confounders: We did not measure nutritional status, specific vitamin levels (particularly vitamin K), or dietary factors that could influence coagulation parameters. This precludes definitive conclusions about the role of malnutrition in the observed PT prolongation.

2. Medication Data: While we collected data on metformin and aspirin use, detailed information on dosage, duration, and adherence was not available.

3. Study Design: As a case-control study, we cannot establish temporal relationships or causality between diabetes and the observed coagulation changes.

                                                                                          

4. Generalizability: Participants were recruited from tertiary centers in Aden, which may not represent the broader Yemeni population or those without access to healthcare.

5. Laboratory Constraints: While we used standardized equipment, reagent scarcity and infrastructure challenges may have introduced measurement variability.

6. Self-reported Data: Some clinical information relied on patient recall and medical records rather than standardized assessments. Future studies should include direct nutritional assessments, measurement of specific coagulation factors and vitamin levels, and prospective designs to establish causal relationships.

## 5. Conclusion

In conclusion, this study provides the first comprehensive characterization of hemostatic parameters in Yemeni patients with T2DM, revealing distinct coagulation abnormalities characterized by prolonged PT and shortened APTT, alongside significant platelet activation that is moderately correlated with glycemic control. These findings highlight observed associations between glycemic control and hemostatic alterations in this patient population and underscore the potential importance of routine coagulation monitoring in diabetic management.

Based on our findings, we recommend the integration of basic coagulation screening (PT/APTT) and platelet indices (MPV) into standard diabetic care protocols in this setting. This approach can help identify patients at increased thrombotic risk and guide appropriate management strategies. The persistence of these coagulation abnormalities after adjustment for multiple confounders suggests a multifactorial pathophysiology that warrants further investigation.

Future studies should include a direct assessment of nutritional status, specific coagulation factors, and micronutrient levels to better elucidate the mechanisms underlying the observed coagulation profile. Prospective designs with clinical outcome tracking would further clarify the prognostic significance of these hemostatic alterations in T2DM patients.

## Supporting information

**S1 File. Data Collection Questionnaire.** Structured questionnaire used to collect demographic and clinical data.
(DOCX)

**S2 File. Contextual Challenges Appendix.** Detailed description of the logistical and infrastructural challenges faced during the study, including electricity supply issues, reagent scarcity, and healthcare access limitations.
(DOCX)

**S3 File. Statistical Analysis Report.** Detailed report of the statistical methods and tests performed, including normality testing, comparative analyses, correlation analyses, and multivariable regression models.
(DOCX)

**S1 Checklist. STROBE checklist.** This checklist is adapted from the STROBE Statement and is licensed under a Creative Commons Attribution 4.0 International license (https://creativecommons.org/licenses/by/4.0/). Original source: https://www.strobe-statement.org/.
(DOCX)

## Acknowledgments

The authors extend their deepest gratitude to the staff of the National Center of Public Health Laboratories and Aden Charity Hospital for their unwavering support and dedication under extraordinarily difficult circumstances. We also thank the patients who participated in this study.

## Author contributions

**Conceptualization:** Naif Taleb Ali.

**Data curation:** Naif Taleb Ali, Gamila Saleh Ali.

**Formal analysis:** Naif Taleb Ali, Gamila Saleh Ali.

**Funding acquisition:** Naif Taleb Ali.

**Investigation:** Naif Taleb Ali, Radfan Saleh Abdullah, Gamila Saleh Ali.

**Methodology:** Naif Taleb Ali.

**Project administration:** Naif Taleb Ali.

**Resources:** Naif Taleb Ali, Radfan Saleh Abdullah, Mansour Abdelnabi H. Mahdi.

**Software:** Naif Taleb Ali, Mansour Abdelnabi H. Mahdi.

**Supervision:** Naif Taleb Ali, Mansour Abdelnabi H. Mahdi.

**Validation:** Naif Taleb Ali, Mansour Abdelnabi H. Mahdi.

**Visualization:** Naif Taleb Ali.

**Writing – original draft:** Naif Taleb Ali, Gamila Saleh Ali.

**Writing – review & editing:** Naif Taleb Ali, Radfan Saleh Abdullah.

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
