## [Decision Letter · Decision Letter 0]

28 Oct 2025

PGPH-D-25-02423

The Paradox of War and Diabetes: Prolonged Prothrombin Time, Shortened APTT, and Platelet Activation in Yemeni Patients with Type 2 Diabetes Mellitus

Dear Dr. Ali,

Thank you for submitting your manuscript to PLOS Global Public Health. After careful consideration, we feel that it has merit but does not fully meet PLOS Global Public Health’s publication criteria as it currently stands. Therefore, we invite you to submit a revised version of the manuscript that addresses the points raised during the review process.

We look forward to receiving your revised manuscript.

Kind regards,

Paolo Ivo Cavoretto, MD PhD

Academic Editor

Journal Requirements:

1. Please ensure that your Ethics Statement is available in its entirety at the beginning of your Methods section, under a subheading 'Ethics Statement'.

2. Please provide separate figure files in .tif or .eps format.

4. We have noticed that you have uploaded Supporting Information files, but you have not included a list of legends. Please add a full list of legends for your Supporting Information files after the references list.

5. In the online submission form, you indicated that your data will be submitted to a repository upon acceptance. We strongly recommend all authors deposit their data before acceptance, as the process can be lengthy and hold up publication timelines. Please note that, though access restrictions are acceptable now, your entire minimal dataset will need to be made freely accessible if your manuscript is accepted for publication. This policy applies to all data except where public deposition would breach compliance with the protocol approved by your research ethics board. If you are unable to adhere to our open data policy, please kindly revise your statement to explain your reasoning and we will seek the editor's input on an exemption.

Additional Editor Comments (if provided):

The external peer review recognized some strenght but required a major revision. Please the authors revise accordingly and resubmit the manuscript for reassessment.

Reviewers' comments:

Reviewer's Responses to Questions

**Comments to the Author**

1. Does this manuscript meet PLOS Global Public Health’s publication criteria?

Reviewer #1: Yes

Reviewer #2: Partly

2. Has the statistical analysis been performed appropriately and rigorously?

Reviewer #1: Yes

Reviewer #2: No

3. Have the authors made all data underlying the findings in their manuscript fully available (please refer to the Data Availability Statement at the start of the manuscript PDF file)?

Reviewer #1: Yes

Reviewer #2: No

4. Is the manuscript presented in an intelligible fashion and written in standard English?

Reviewer #1: Yes

Reviewer #2: Yes

Reviewer #1: 1-The phrase "The Paradox of War" in the title is intended to highlight the unique and complex interplay between the conflict environment and the observed hemostatic abnormalities in Yemeni patients with T2DM. It emphasizes how war-related factors—such as malnutrition, healthcare system collapse, and chronic stress may influence or exacerbate the paradoxical coagulation profile (prolonged PT with shortened APTT). The following are Suggestions for Improved Titles:

• Hemostatic Alterations in Yemeni Patients with Type 2 Diabetes Mellitus Amid Conflict Conditions

• Coagulation and Platelet Activation in Yemeni T2DM Patients: Impact of Conflict and Malnutrition

• Paradoxical Coagulation Profiles in Yemeni T2DM Patients: Influence of War and Resource Scarcity

• Hemostatic Dysfunction in Yemeni Patients with T2DM: Effects of Conflict-Related Malnutrition

• The Impact of Conflict on Coagulation and Platelet Activation in Yemeni T2DM Patients"

• Altered Coagulation and Platelet Indices in Yemeni Patients with T2DM: A Conflict-Affected Population

2-Include recent studies on coagulation abnormalities and platelet indices in T2DM patients from different regions, emphasizing similarities and differences. Highlight regional data from the Middle East or conflict zones to contextualize your findings, please add the following recent study from middle est.

• Role of platelet indices and vitamin D in forecasting deterioration of glycemic control and vascular complications in type 2 diabetes. Vol. 19 No. 1 (2025). Italian Journal of Medicine. https://doi.org/10.4081/itjm.2025.1875

3-Provide more insights into how conflict-related stress, malnutrition, and healthcare limitations specifically influence coagulation pathways beyond laboratory results.

4-Support your observation of prolonged PT and shortened APTT with previous literature reporting similar or contrasting patterns, analyzing possible reasons (e.g., vitamin deficiencies, inflammation)

5-Elaborate on potential mechanisms behind the paradoxical coagulation profile, referencing studies on vitamin K deficiency, hepatic function, and inflammatory cytokines.Discuss how conflict-related factors like malnutrition and drug access barriers might modify coagulation differently than in stable populations.

6-Clearly acknowledge limitations like the cross-sectional design, lack of specific coagulation factor testing, and generalizability. Suggest future longitudinal or mechanistic studies, including the measurement of coagulation factors and micronutrient levels.

Reviewer #2: Review Summary

The manuscript is generally well articulated and structured as expected of observational studies in clinical epidemiology. The authors exhibited good knowledge of the subject matter, and the findings has potential to contribute new understanding of the hypercoagulation profile of type-2 diabetic patients in the study setting, which may better inform the global efforts to control this increasingly challenging metabolic disease in similar context. Contrary to the sensational title that highlights war and the authors’ narrative fixated on potential effects of war-induced malnutrition and disruptive access to essential healthcare services in a typical humanitarian setting, this manuscript did not report any assessment of variables that measures nutritional status, use or access to needed medications or treatment in the diabetic and non-diabetic Yemeni participants studied. Instead of speculative narration of what could have been, the authors had the luxury of measuring these important confounding nutritional and healthcare access factors of interest, adjust their analysis for them, in order to correctly make any valid allusions to their possible effects on the observed prolongation of PT in this study.

Unfortunately, nutritional status was not measured in this study, use of metformin that could alter bioavailability of vitamin k needed for endogenous synthesis of essential coagulation factors was also not measured. So, it is spurious to insinuate that every person in Yemen (by extension, the 240 persons studied at Aden) is suffering from inherent malnutrition by default because Yemen is a known conflict setting. No evidence in this manuscript shows malnutrition in the participants, especially undernutrition, and no variables was included to show whether they were using metformin or not. Hence, it is concerning that the un-adjusted atypical finding of prolonged average PT in the type-2 diabetic participants is being speculatively explained by the authors as some novel paradox due to war induced-malnutrition, when a simple stratified sub-analysis or multiple regression could reject or confirm that reasoning. It seems illogical to draw attention to very important confounding factors that could have been easily measured but were not measured, when such should be some anecdotal notes in the sub-section for study setting or discussion of study limitations.

However, this study clearly assessed some selected coagulation or haemostatic markers in a group of 140 hospital-based type-2 diabetes mellitus (T2DM) patients in Yemen, attempted to determine the crude effect of their glycaemic control status on these markers, and compared these findings with 100 age-sex-matched non-diabetic participants. The authors were not sure if they conducted a cross-sectional or a case-control study as they reported both as the study design. This is an unacceptable methodological anomaly which may be reflective of the inefficient use of more appropriate statistical techniques in their analysis, poor interpretation of their findings, and the inconsistent use of applicable technical terminologies (e.g. cases, controls, cohort, group, patients, participants, etc) in their narrative across the manuscript.Based on the highlighted technical concerns, it is apparent that the study design, analysis and narrative will benefit from substantial modification. Accordingly, I am recommending a major revision of this manuscript before its acceptance for publication.

Details of the Issues in the Manuscript

To support the authors’ revision efforts, it is hereby suggested that the authors should revise the manuscript to address the following highlighted concerns:

1) Title & Abstract

As currently written, the obvious paradox in this manuscript is the author’s decision to leave out the measurement and adjustment for clearly important confounding factors of interest, which could have explained the observed prolongation of average PT among the diabetic patients. If the authors’ wish to make this manuscript about war-induced crippling of the healthcare system and widespread malnutrition, it is basic that they measure variables that assesses these factors and adjust their analysis for them. Another paradox is the anomaly of defining the study design as cross-sectional in the abstract but, it is stated as cross-sectional and case-control in the main text. Also, who were the diabetic participants compared with to arrive at the claim of significance in the observed poor glycaemic control (86%) as reported under the result sub-section of the abstract? The adjectival term “significant” is inappropriate to express a descriptive summary of a variable. In its statistical or clinical meaning, a comparison is required to assert a significant difference. Lastly, the conclusive statement is more aspirational than anything that can be specifically deduced from the index study. In this study, participants who were currently using anti-coagulants or anti-platelets medications were excluded. Therefore, since the observed average PT of the diabetic participants were already prolonged in this study, why will this finding support any advocacy for more anti-thrombotic therapy in conflict settings when such will likely worsen the already prolonged PT? Accordingly, the title and abstract should be re-written to reflect what the real study objective, methods and findings were.

2) Introduction and Study Objectives

The authors argued in the second paragraph of the introduction that a prolonged PT is one of the typical markers seen among diabetic patients. First, this is not truly affirmed by the literature referenced. Secondly, if that were to be taken as is, the prolonged PT seen in the index study result would not be a paradox or abnormal finding to highlight. More correctly, hypercoagulability is NOT an inevitable marker of diabetes in all patients living with the disease. Rather, it is a common marker of “uncontrolled diabetes” that is directly explained by poor glycaemic control, where a shortened PT is the commonest finding but, a prolonged PT has been reported, as well. Again, these further questions the proclamation of some new paradox associated with war by this manuscript. The authors should revamp their argument to be more coherent and consistent with existing literature and their research questions.

In the 3rd paragraph, the authors highlighted that Yemen is experiencing a rising prevalence of T2DM. This is consistent with global trend in virtually every country, with or without conflict or any disruption of access to healthcare services. Why are the authors still pushing the narrative of a possible conflict effect or explanation for a demographic transition and lifestyle change issue? More so, the participants in this study were recruited at tertiary-care diabetic clinic, implying that they had access to specialist diabetic care. A population-based study that included participants with no access to care or diagnosis might justify the generalization of no access to healthcare. Variables measuring degree of access to appropriate care and treatment must be measured and adjusted for to show if conflict-induced disruption of healthcare had any meaningful effect on the coagulation profile of the hospital-based participants studied.

In the 4th an 5th paragraph, the study aims were unnecessarily repeated without any added information. More importantly, this study did not report what can be defined as comprehensive coagulation parameters and platelet indices. The reported assessment of PT, APTT, MPV and PDW can best be labelled a selection of key haemostatic or coagulation indicators. Also, it may be more correct to state that this study simply assessed the crude or unadjusted association of selected coagulation parameters with glycaemic control status among type-2 diabetic and non-diabetic adults in Aden, Yemen. The authors should consider modifying the introduction and study objectives to highlight more relevant information with improved specificity to what was actually measured and studied.

3) Methods

A study design is literally a technical box that guides what can be done, how it is done and what can be inferred from the findings of a given research endeavour. Therefore, the authors cannot be ambivalent about what study design was used in this study. A cross-sectional study, even when analytic, is very different from a case-control study. It is comical for the authors to claim they did a cross-sectional, case-control study. This is a fundamental assertion that ridicules the research methodological tenacity of this manuscript. The authors need to state the specific epidemiological study design that was applied in the conduction of this study, which is the fulcrum around which every other aspect of their methodological approach and interpretation of the findings must revolve.

What is stated as contextual challenges under sub-section 2.2 will be best left as discursive points when highlighting limitations of methods and findings in the appropriate paragraphs under the discussion section. In any case, the authors need to be reminded that these general challenges cannot be summarily adopted to have affected this study if they cannot highlight exactly how each of the challenge limited the accuracy and precision of their methodological processes including the recruitment of study participants, assessment of variables of interest and the analysis of results. For example, how did the use of back-up generator change anything about this study? They said laboratory reagents were scarce but, if they got the right reagents and used them correctly, how did reagent scarcity affect the findings of this study? Also, the authors stated that “patient access to consistent diabetic care and anticoagulant therapy was limited, as reported in our cohort, where less than 15% were on regular aspirin”. I failed to find any variable in any aspect of this manuscript that reported this. So, what cohort are they referring to? Moreso, how is this a challenge to this study when in the following sub-section, one of the exclusion criteria was “use of anticoagulant or antiplatelet medication”? Lastly, the authors boldly declared that: “These contextual factors are not just background, but active determinants of the hemostatic profile observed, reflecting a reality where disease pathophysiology is inextricably linked to systemic healthcare failure”. For the umpteenth time, as I have highlighted repeatedly, if these factors are perceived to be important determinant of haemostatic profile, then they should have been defined as confounding variables, measured, included in the analysis and adjusted for. However, some of the information about contextual challenges can be better re-phrased under study setting.

Concerning sub-section 2.3, sample size estimation formulars are study design specific. With the authors’ ambivalence about the specific epidemiologic study design applied, the correctness of the quoted minimum sample size is questionable. Howbeit, no reference was provided for the goal to detect a clinically significant difference in MPV of 1.0fl. Is this an arbitrary choice? Ideally, a minimum sample size should be estimated for each main independent variable of interest, and the largest minimum will be adopted as the rational choice as the minimum sample size, to ensure the studied number of participants are adequate to detect the anticipated difference for all key variables. MPV is only one of many other factors of interest in this study, is there any reason for the choice of MPV as the sample size estimation variable? More concerning is the fact that MPV is one of the outcome variables in this study, as portrayed. The variable(s) to be used to estimate sample size should the predictor (independent) variables that you wish to estimate their effect(s) on the outcome. Also, the authors will need to clearly specify and define the outcome (dependent) and predictor (independent) variables for this study under an appropriate sub-heading. Assuming the estimated sample size is correct, 120 participants were needed in each group. Statistically, increasing the number of non-diabetic (healthy controls) is justifiable but, ethically, it is unusual to increasing the number of diabetic (ill persons) while reducing the number of controls. The authors quickly claimed “well-documented constraints of the research setting” for this but, they need to be more specific about what constraint made it easier to find more type 2 diabetic patients than healthy controls for this study in Aden.

About the study participants, the 140 type-2 diabetic patients were reportedly enlisted from an outpatient clinic but, the use of a simple random sampling technique in require a few more details. To claim simple random sampling was truly done, the authors are affirming that they had a sampling frame and line-list of the total patients. So, they should report the total population size of the type-2 DM patients attending that clinic, and how they randomly selected 140. The non-diabetic participants, whom the authors tagged “??healthy controls”, seemed to have qualified as “healthy” by their personal medical history profile alone. Are the authors saying they relied on the validity of self-reported medical history to ascertain if they were healthy, free of type-2 diabetes and other diseases? The least requirement should be some clinical assessment by a licensed physician if other screening investigations are two expensive for the research budget. Also, the post-hoc fasting glucose levels shown in their results table-1 could suffice as proof that they were not diabetic. More importantly, the authors were silent about the source of these “healthy controls”. Were they other patients attending the same hospital for other reasons or some wider community members? How were they selected? Did they apply simple random sampling technique, too? Knowing the study-base for the control group is a very important methodological information. The general inclusion and exclusion criteria that applied to the diabetic group should be applied to the non-diabetic (healthy) group, except that the former should be diagnosed of type-2 diabetes.

I will suggest that sub-section 2.5 should be expanded to define the key variables measured in this study, so that it can be abundantly clear whether your use of those terms is the same or different from the literature definitions. It is also the right place to highlight the outcome, main predictor(s) and control variables. Information about blood sample collection can be moved under laboratory procedure or technique (subsection2.6). Finally, the statistical analysis narrative is generic and will certainly need to be re-written, after the authors have chosen a specific study design and defined their variables clearly. Irrespective of whether they did a cross-sectional or a case-control study, the effect of glycaemic control on the observed haemostatic markers will require adjustment for confounding variables by stratified sub-analysis or multiple regression.

4) Results, Discussion and Conclusion

Invariably, a new statistical analysis will be required in the process of revising this manuscript. Hence, the entirety of the results, discussion and conclusion will require re-writing after all the already highlighted issues in concept, methods and analysis are addressed.

5) References, Tables & Figures

The numbered references can be better organised and presented, as some of the listed of references lacks appropriate content required by standard referencing styles.

Table-1 and Table-2 do need to be separated. Both tables simply show the characteristics of the participants. Authors need to clarify if Figure-2 included all participants or a subset. Figure-3 will be best placed side-by-side with the line graph for the non-diabetic group. All the figures should be better modified, improved and re-presented, as necessary.

**Do you want your identity to be public for this peer review?** For information about this choice, including consent withdrawal, please see our Privacy Policy

Reviewer #1: No

Reviewer #2: **Yes: ** Abraham Braimah Idokoko

---

## [Decision Letter · Decision Letter 1]

25 Nov 2025

PGPH-D-25-02423R1

Altered Coagulation and Platelet Indices in Yemeni Patients with Type 2 Diabetes Mellitus: A Conflict-Affected Population

Dear Dr. Ali,

Thank you for submitting your manuscript to PLOS Global Public Health. After careful consideration, we feel that it has merit but does not fully meet PLOS Global Public Health’s publication criteria as it currently stands. Therefore, we invite you to submit a revised version of the manuscript that addresses the points raised during the review process.

We look forward to receiving your revised manuscript.

Kind regards,

Paolo Ivo Cavoretto, MD PhD

Academic Editor

Journal Requirements:

Additional Editor Comments (if provided):

Reviewer 1 was satisfied by the revision while reviewer 2 still raises major concerns on methodology and the editor approves the need for a further round of revision. Please address ALL comments of the referee carefully and resubmit the paper for reassessment.

Reviewers' comments:

Reviewer's Responses to Questions

**Comments to the Author**

Reviewer #1: All comments have been addressed

Reviewer #2: (No Response)

publication criteria?

Reviewer #1: Yes

Reviewer #2: Partly

3. Has the statistical analysis been performed appropriately and rigorously?

Reviewer #1: Yes

Reviewer #2: No

4. Have the authors made all data underlying the findings in their manuscript fully available (please refer to the Data Availability Statement at the start of the manuscript PDF file)?

Reviewer #1: Yes

Reviewer #2: No

5. Is the manuscript presented in an intelligible fashion and written in standard English?

Reviewer #1: Yes

Reviewer #2: Yes

Reviewer #1: Thank you for addressing all of my suggestions, there is minor correction in references no 10, the author name are not belong to the references, see the correct citation as follow:

Obaid Al Ali, Maryam Ahmed, Khalid Abdelsamea Mohamedahmed, and Asaad Ma Babker. "Role of platelet indices and vitamin D in forecasting deterioration of glycemic control and vascular complications in type 2 diabetes." Italian Journal of Medicine 19, no. 1 (2025).

Reviewer #2: Review of the Revised manuscript Version (PGPH-D-25-02423R1)

Primarily, I wish to commend the author’s prompt submission of their revised manuscript, with substantial changes made but, the following issues on the draft document will require some more attention before acceptance for publication:

1) Title, Abstract & Introduction

Generally, the authors should be reminded that they are writing a scientific paper, and not some opinion piece. In other words, their textual narrative should be evidently supported by the numbers. More importantly, the statistical procedures generating the numbers should be correctly done to make the numbers credible to the potential users of their hard-earned research output. For example, the 0.52 correlation coefficient reported under result in the abstract is NOT considered “strong positive correlation” by any classification system. That is moderate correlation, at best. So, it will be more helpful to their hard work to honestly interpret their methods and findings, instead of the needless exaggerations, and sensational adjectives. This is a major concern that apply to every aspect of the methods, results and discussion sections of this manuscript.

2) Methods, Results and Discussion

The authors have corrected their study design as a case control study, with useful adjustment. As previously advised, their reporting attempt will be more optimized if they consult the STROBE guidelines and follow its recommendations.

Strikingly, the authors still did not define their main outcome(s) of interest, primary predictor(s), such that we can consider all other variables as covariates. Doing this will enhance their attempt at using appropriate statistical techniques and interpreting results more correctly, especially the multiple regression modelling. For example, the study objective is ambivalent about whether glycaemic level is the outcome or it is coagulation or platelet parameters. This should be clarified in the definition of variables in the methods section so that it can be clear what specific outcome(s) is being modelled, the main predictor(s) that we are estimating its effect on the outcome, such that other covariates are adjusted for their confounding, mediating or modifying effects.

If coagulation and platelet parameters are the outcomes of interest by the consistent narrative in the manuscript text, why are the authors reporting beta-coefficients and p-values for their effects, as shown in table 3? What outcome was this coagulation parameters having effects on? It will be counter-intuitive if the authors were trying to predict glycaemic levels with coagulation parameters in their model. Why? How? What is the logic?

Also, there are only eight measured variables reported in this study, as shown in tables 1 and table 2. These are few enough for the authors to be less generic about their statistical technique for handling each variable. After the previous review recommendations, the authors claimed to have adjusted their multiple regression model for age, sex, metformin use, hypertension, and cardiovascular history. The age and sex distribution of the participants are shown in table 1 but, I could not find the numbers for the other three variables. If they measured and adjusted for these variables, it is more credible to show their descriptive numbers, even if they wish to leave these covariates out of the multiple regression model table 3.

“Multivariable regression” is a broad theme that is often used to describe a large group of simple and multiple regression procedures that apply to different “univariate or multivariate OUTCOME variable types” with strictly different dataset terms and conditions. I hope the authors can stick to using the specific term “multiple linear regression” that was conducted in this study in their narrative, rather than “multivariable regression”. Also, the authors made three models for unclear reasons. I could decipher the adjustment made in model 2 and model 3, as stated. What does the “unadjusted model 1” contain? There should be at least one outcome and predictor in it. State them. Lastly, I will advise the optional reporting of the effect sizes for the covariates included in the multiple regression model, as they are of interpretative value, as well as the main predictors.

Finally, the enormous work put into this research will be most rewarding to all stakeholders if the authors will consider meticulous attention to their methodology, statistical procedures and correct interpretation. We all know that findings and conclusions of research are ONLY as valid or credible as the methods and statistical techniques applied. The authors should take another look at my previous review recommendations for other pending concerns.

**Do you want your identity to be public for this peer review?** For information about this choice, including consent withdrawal, please see our Privacy Policy

Reviewer #1: No

Reviewer #2: **Yes: ** Abraham Braimah Idokoko

---

## [Editor Report · Decision Letter 2]

2 Dec 2025

PGPH-D-25-02423R2

Altered Coagulation and Platelet Indices in Yemeni Patients with Type 2 Diabetes Mellitus: A Conflict-Affected Population

Dear Dr. Ali,

Thank you for submitting your manuscript to PLOS Global Public Health. After careful consideration, we feel that it has merit but does not fully meet PLOS Global Public Health’s publication criteria as it currently stands. Therefore, we invite you to submit a revised version of the manuscript that addresses the points raised during the review process.

We look forward to receiving your revised manuscript.

Kind regards,

Paolo Ivo Cavoretto, MD PhD

Academic Editor

Journal Requirements:

Additional Editor Comments:

There were improvements but not all issues were resolved. As pointed out by the referees in the previous round the authors need to revise some statements:

Abstract: "A strong positive correlation was observed between HbA1c and MPV (r = 0.52, p < 0.001)." This is moderate correlation at best, please revise in the abstract as well as in the mein text and conclusions.

Please smooth the conclusion as association does not imply causation.

There are insististencies in the manuscript: the chapter Sample Size Justification is written twice (pag 5) please revise.
---

## [Editor Report · Decision Letter 3]

10 Dec 2025

Altered Coagulation and Platelet Indices in Yemeni Patients with Type 2 Diabetes Mellitus: A Conflict-Affected Population

PGPH-D-25-02423R3

Dear Dr Ali,

We are pleased to inform you that your manuscript 'Altered Coagulation and Platelet Indices in Yemeni Patients with Type 2 Diabetes Mellitus: A Conflict-Affected Population' has been provisionally accepted for publication in PLOS Global Public Health.

Best regards,

Paolo Ivo Cavoretto, MD PhD

Academic Editor